# Latent Tuberculosis Infection and COVID-19: Analysis of a Cohort of Patients from Careggi University Hospital (Florence, Italy)

Francesca Mariotti [1,*]🆔, Francesco Sponchiado [1], Filippo Lagi [2], Chiara Moroni [3], Riccardo Paggi [1], Seble Tekle Kiros [1]🆔, Vittorio Miele [3], Alessandro Bartoloni [1,2]🆔, Jessica Mencarini [2] and The COCORA Working Group †

1 Department of Experimental and Clinical Medicine, University of Florence, 50134 Florence, Italy; riccardo.paggi@unifi.it (R.P.); sebletekle.kiros@unifi.it (S.T.K.)
2 Infectious and Tropical Diseases Unit, Careggi University Hospital, 50134 Florence, Italy; mencarinij@aou-careggi.toscana.it (J.M.)
3 Emergency Radiology Unit, Careggi University Hospital, 50134 Florence, Italy
* Correspondence: francesca.mariotti@unifi.it
† COCORA working group members are listed at the end of the article.

**Abstract:** Data regarding the relationship between coronavirus disease (COVID-19) and active or latent tuberculosis (TB) are discordant. We conducted a retrospective study examining the impact of latent tuberculosis infection (LTBI) on the clinical progression of COVID-19 patients. We selected 213 patients admitted with COVID-19 in a tertiary-level Italian hospital (February–December 2020), who underwent a QuantiFERON-TB test (QFT) and/or chest radiological exam. The population was divided into three groups: (i) QFT negative and without radiological TB sequelae (Neg); (ii) QFT positive and without radiological TB sequelae (Pos); (iii) radiological TB sequelae regardless of QFT result (Seq). In-hospital mortality and oro-tracheal intubation (OTI) showed significantly higher results in the Seq group (Seq 50% vs. Pos 13.3% vs. Neg 9.3%, $p < 0.001$; Seq 16.7% vs. Pos 6.7% vs. Neg 4.9%, $p = 0.045$). Considering the Pos and Seq groups' patients as the population with defined LTBI, in-hospital mortality (20/51, 39.2%) and OTI risk (7/51, 13.7%) were statistically higher with respect to patients without LTBI (in-hospital mortality: 15/162, 9.3%, $p < 0.001$; OTI risk: 8/162, 4.9%, $p = 0.023$), respectively. Multivariate analysis showed that radiological sequelae and the Charlson Comorbidity Index (CCI) were significantly associated with higher mortality rate; despite the higher CCI of Seq population, we cannot exclude the correlation between COVID-19 in-hospital mortality and the presence of radiological TB sequelae.

**Keywords:** latent tuberculosis infection; COVID-19; QuantiFERON-TB test; tuberculosis radiological sequelae; in-hospital mortality





## 1. Introduction

The coronavirus disease (COVID-19) pandemic has become a public health emergency because of its exponential expansion worldwide and how it considerably impacts morbidity and mortality. Since December 2019, the COVID-19 global burden is over 771 million cases and 6.9 million deaths as reported by the World Health Organization (WHO) [1]. Until the COVID-19 pandemic, tuberculosis (TB) was the leading cause of death from a single infectious agent with a total of 1.4 million deaths and 10 million cases in 2019 worldwide. Even today, the COVID-19 pandemic continues to negatively impact TB diagnosis and treatment, slowing the progress made in previous years due to global TB elimination programs, and thus increasing the burden of TB disease [2].

Latent tuberculosis infection (LTBI) is defined as a state of persistent immune response to *Mycobacterium tuberculosis* (MT) antigens with no evidence of clinically manifested TB.

Approximately a quarter of the world's population is estimated to be infected with TB. Active TB may develop in 5 to 10% of people with latent infection during their lifetimes, usually within the first 5 years. Different factors can lead to cellular immunity suppression, influencing the likelihood of progressing to active TB disease, such as human immunodeficiency virus infection, tumor necrosis factor $\alpha$ inhibitors and glucocorticoids use, organ or hematologic transplantation, diabetes, and end-stage renal disease [3–5]. Currently, there are no microbiological diagnostic tests to detect LTBI; tuberculin skin test (TST) and interferon-$\gamma$ release assays (IGRAs), comprehensive of QuantiFERON-TB Gold In-Tube assay (QFT) and T-SPOT.TB assay indirectly reveal TB infection by detecting memory T-cell response to MT antigens [6]. Chest radiography (CXR) and computed tomography (CT) are mandatory to distinguish LTBI from active disease, particularly since negative IGRA or TST cannot exclude active TB or rule out LTBI in immunocompromised patients. Moreover, in selected individuals with a high risk of TB reactivation with a positive TST/IGRA and inconclusive CXR findings, a chest CT can help to distinguish between active and latent disease [7,8].

In the available literature, data regarding the relationship between COVID-19 and active or latent TB are few and discordant. Some reports from TB endemic countries evidenced that COVID-19 patients coinfected with TB (COVID-TB) have a higher risk of death than those infected with a single pathogen [9,10], leading to a high mortality rate, especially in the elderly population [11–14]. Conversely, an Italian study reported a lower mortality rate in COVID-TB patients, with results likely being influenced by a younger population and fewer comorbidities [15].

We present a study to examine the impact of patients with LTBI on COVID-19 clinical progression and in-hospital mortality in a pre-vaccinal era and in a low-TB prevalence area.

## 2. Materials and Methods

### 2.1. Patient Population and Study Design

We retrospectively retrieved data on patients admitted for COVID-19 in Tropical and Infectious Diseases and three Internal Medicine wards of Careggi University Hospital, Florence, Italy, between 1 February 2020 and 31 December 2020. Patients were considered eligible if: (i) they were ≥18 years old; (ii) they had a confirmed SARS-CoV-2 positivity by polymerase chain reaction on a nasopharyngeal swab or bronchoalveolar lavage; (iii) they were hospitalized for COVID-19.

Patients were admitted and treated for COVID-19 according to COVID-19 management guidelines [16,17]. In case of worsening clinical features, patients performed radiological (e.g., CXR, chest CT) and microbiological (e.g., blood culture, bronchoalveolar lavage) exams to exclude the presence of other concomitant pathologies (e.g., active TB, *Pneumocystis jirovecii* pneumonia, aspergillosis).

Radiological images (CXR and chest CT) available in the patient clinical file and performed prior (up to 2 years before) or during the hospitalization were considered and reviewed by two thoracic-trained radiology specialists and a clinician from our hospital, unaware of QFT result. According to the literature [18], we defined TB sequelae as the presence of lung apices lesions and/or perilary lymphnode sites calcification. Chest radiological images with massive COVID-19 pneumonia were excluded since TB sequelae were not clearly distinguishable.

A QuantiFERON-TB test was performed to allow the use of immunosuppressive drugs during the hospitalization. We used QFT cut-off values of the Reference Regional Centre for Tropical Diseases of Careggi University Hospital, Florence (positive if CD4 and CD4/CD8 ≥ 0.35 UI/mL). We defined LTBI in positive cases of QFT and/or the presence of radiological sequelae.

Based on QFT results and radiological findings, the population was divided into three groups: (i) QFT negative and without radiological TB sequelae (Neg); (ii) QFT positive and without radiological TB sequelae (Pos); (iii) radiological TB sequelae regardless of QFT result (Seq).

Patients were excluded if: (i) they had QFT indeterminate result; (ii) they had not performed QFT nor radiological examinations with the characteristics described above; or (iii) they had unknown hospital admission dates for lack of clinical data.

### 2.2. Data Collection

We analyzed patient's data using electronic medical records available concerning demographic (sex, age, region of birth), clinical and hospitalization characteristics (comorbidities, Charlson Comorbidity Index (CCI), Horowitz Index, immunosuppressive therapy before hospitalization, COVID-19 treatment, non-invasive ventilation support, intensive care unit (ICU) admission, oro-tracheal intubation (OTI), and in-hospital mortality). We collected immune, microbiological, and radiological test results to diagnose LTBI and COVID-19. The collected anonymized data were subsequently entered into a computerized database (.xlsx file, Microsoft Excel 365®).

### 2.3. Outcomes

The primary outcomes were in-hospital mortality and OTI risk due to COVID-19 progression. A composite outcome consisting of OTI and in-hospital mortality was also considered.

### 2.4. Statistical Analysis

Descriptive analysis was employed to illustrate population characteristics. Categorical variables were evaluated with X2/Fisher's exact test, while continuous variables were assessed with the Mann–Whitney test when two groups were compared, or the Kruskall–Wallis test if three groups were compared. A multivariate analysis by Cox regression was used to examine the association between death and selected variables (CCI, belonging to Pos and Seq groups, sex, previous therapy with immunosuppressive drugs, therapies for COVID-19). STATA v18.0 (STATACorp, College Station, TX, USA) was used for statistical analyses. A *p* value less than 0.05 was considered statistically significant.

## 3. Results

During the study period, 495 patients were admitted for COVID-19 to Careggi University Hospital.

Of the 495 patients, QFT was known in 202 (40.8%): 169 of them (83.7%) were negative, 22 (10.9%) positive, and 11 (5.4%) indeterminate. Radiological pictures were available for 135/495 patients (27.3%): in 36 patients, lesions compatible with TB sequelae were evidenced. Excluding patients with indeterminate QFT (11) and without hospitalization dates (9), 213 patients had radiological evidence of TB sequelae and/or a QFT available.

The majority was represented by males (120/213, 56.3%), with a mean age of 61 ± 19 years; 75.1% (160/213) patients were of Italian nationality. Most represented comorbidities were arterial hypertension (99/213, 46.5%), diabetes mellitus (36/213, 16.9%), chronic obstructive pulmonary disease (26/213, 12.2%), coronary artery disease (22/213, 10.3%), and cerebrovascular disease (18/213, 8.5%). Sixty-four (30.0%) patients were characterized by a CCI ≥ 5. Steroid therapy and tocilizumab were administered to 152/213 (71.4%) and 12/213 (5.6%) patients, respectively. Fifty-one patients (51/213, 23.9%) required non-invasive ventilation, 33/213 (15.5%) were transferred to ICU, and 15/213 (7.0%) underwent OTI.

Of 213 patients, 162 (76.1%) were assigned to Neg group, 15 (7.0%) to Pos group, and 36 (16.9%) to Seq group. General characteristics divided by group are summarized in Table 1. The three groups were significantly different in terms of age and CCI. Italian nationality was significantly prevalent in Seq group with respect to the others (Seq 88.9% vs. Pos 53.3% vs. Neg 74.1%, *p* = 0.023). Thirty-five patients (35/213, 16.4%) died during the hospitalization: 15 (42.9%) belonged to Neg, 2 (5.7%) to Pos, and 18 (51.4%) to Seq groups. The risk of in-hospital mortality in Seq group was significantly higher than in Neg and Pos groups, since 18/36 patients belonging to Seq group died (Seq 50% vs. Pos 13.3% vs. Neg

9.3%, *p* < 0.001). The proportion of patients who underwent OTI in Seq cohort was 16.7%, significantly higher with respect to the other cohorts (Pos 6.7% vs. Neg 4.9%, *p* = 0.045). Considering the composite outcome, Seq cohort in-hospital mortality and/or OTI risk was significantly higher (Seq 52.8% vs. Pos 13.3% vs. Neg: 13.6%, *p* < 0.001). Primary and composite outcomes in different groups are resumed in Table 1.

**Table 1.** General and clinical characteristics of study population, divided into patients with negative QFT, positive QFT, and radiological findings of TB sequelae.

| Characteristics | Negative (n = 162) | Positive (n = 15) | Sequelae (n = 36) | *p*-Value |
|---|---|---|---|---|
| Male (n, %) | 86 (53.1) | 9 (60.0) | 25 (69.4) | *p* = 0.193 |
| Age (mean ± SD) | 61.3 ± 18.9 | 62.7 ± 20.9 | 75.8 ± 13.0 | ***p* < 0.001** |
| Italian nationality (n, %) | 120 (74.1) | 8 (53.3) | 32 (88.9) | ***p* = 0.023** |
| CCI (mean ± SD) | 2.8 ± 2.5 | 2.7 ± 2.6 | 4.9 ± 2.6 | ***p* < 0.001** |
| IT before hospital admission (n, %) | 11 (6.8) | 2 (13.3) | 4 (11.1) | *p* = 0.502 |
| Steroid therapy (n, %) | 116 (71.6) | 11 (73.3) | 25 (69.4) | *p* = 0.952 |
| Tocilizumab therapy (n, %) | 9 (5.6) | 1 (6.7) | 2 (5.6) | *p* = 0.984 |
| ICU admission (n, %) | 23 (14.2) | 4 (26.7) | 6 (16.7) | *p* = 0.425 |
| In-hospital deaths (n, %) | 15 (9.3) | 2 (13.3) | 18 (50) | ***p* < 0.001** |
| OTI | 8 (4.9) | 1 (6.7) | 6 (16.7) | ***p* = 0.045** |
| Death and/or OTI (n, %) | 22 (13.6) | 2 (13.3) | 19 (52.8) | ***p* < 0.001** |

Bold character was used for statistically significant *p*-values. CCI: Charlson Comorbidity Index; ICU: intensive care unit; IT: immunosuppressive therapy; SD: standard deviation; QFT: QuantiFERON-TB test; OTI: oro-tracheal intubation.

As showed by Kaplan–Meier survival curves (Figure 1), both in-hospital mortality and composite outcome were significantly higher in Seq group (log rank test, *p* < 0.001).

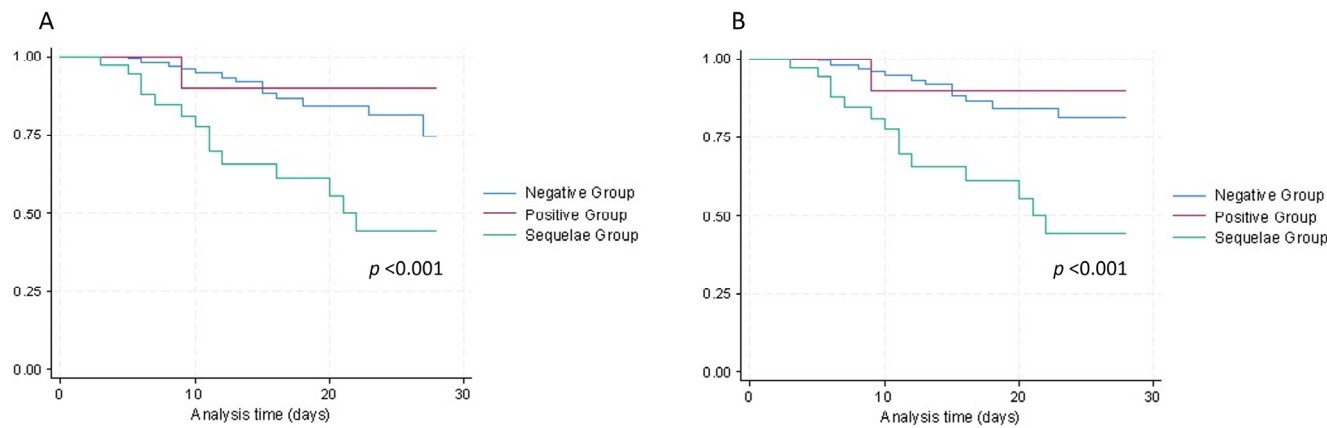

**Figure 1.** Cumulative survival (**A**) and combination of cumulative survival and avoided oro-tracheal intubation (**B**) of COVID-19 admitted patients in a single center in Italy during hospitalization, divided in patients with negative QFT, positive QFT, and radiological findings of TB sequelae.

Considering Pos and Seq groups' patients as a population with defined LTBI, in-hospital mortality (20/51, 39.2%) and OTI risk (7/51, 13.7%) were statistically higher with respect to patients without LTBI (in-hospital mortality: 15/162, 9.3%, *p* < 0.001; OTI risk: 8/162, 4.9%, *p* = 0.023), respectively.

By Cox's regression multivariate analysis, CCI and belonging to Seq group were significantly associated with in-hospital mortality (Hazard Ratio 1.16, 95% CI 1.04–1.30 and Hazard Ratio 3.00, 95% CI 1.34–6.73, respectively). The same variables were not significantly associated with higher OTI risk (Table 2).

**Table 2.** Multivariate analysis by Cox regression used to examine the association between in-hospital mortality/oro-tracheal intubation risk and selected variables (sex, Charlson Comorbidity Index, Pos and Seq groups belonging, immunosuppressive therapy before hospital admission, COVID-19 therapies).

| | HR | *p*-Value | [95% CI] |
|---|---|---|---|
| **In-hospital death** | | | |
| Sex | 1.13 | *p* = 0.755 | 0.53–2.39 |
| CCI | 1.16 | ***p* = 0.007** | 1.04–1.30 |
| Pos group belonging | 1.55 | *p* = 0.567 | 0.35–6.92 |
| Seq group belonging | 3.00 | ***p* = 0.008** | 1.34–6.73 |
| IT before hospital admission | 0.65 | *p* = 0.484 | 0.19–2.20 |
| Steroid therapy | 1.91 | *p* = 0.266 | 0.61–5.98 |
| Tocilizumab therapy | 0.67 | *p* = 0.622 | 0.14–3.23 |
| | | | |
| **OTI** | | | |
| Sex | 2.66 | *p* = 0.220 | 0.56–12.82 |
| CCI | 0.88 | *p* = 0.266 | 0.69–1.11 |
| Pos group belonging | 3.93 | *p* = 0.237 | 0.41–38.06 |
| Seq group belonging | 4.48 | *p* = 0.072 | 0.87–22.99 |
| IT before hospital admission | 6.65 | ***p* = 0.023** | 1.29–34.29 |
| Steroid therapy | 0.72 | *p* = 0.726 | 0.12–4.42 |
| Tocilizumab therapy | 3.72 | *p* = 0.104 | 0.76–18.10 |

Bold character was used for statistically significant *p*-values. CCI: Charlson Comorbidity Index; IT: immuno-suppressive therapy; OTI: oro-tracheal intubation; Pos: group with QFT positive and without radiological TB sequelae; Seq: group with radiological sequelae regardless of QFT result.

## 4. Discussion

COVID-19-associated mortality is notably increased by aging, disabilities, and under-lying medical conditions [14,19–21]. However, data regarding the relationship between COVID-19 outcome and active or latent TB are few and discordant in the available litera-ture. Clinical evidence suggests that COVID-19 may predispose patients to TB disease or reactivation of latent TB infection through severe depletion and dysfunction of T-cells and uncontrolled production of pro-inflammatory cytokines [22].

According to some studies, poorer COVID-19 outcomes could be associated with active TB, probably due to the dysregulation of immunity during MT infection [11,13,14,21,23]. Global TB/COVID-19 network study group evidenced that patients diagnosed with COVID-19 after the end of TB treatment had a poorer prognosis in comparison with those with COVID-19 diagnosed before or during TB treatment [13]. This observation suggests that pre-existing TB lung disease could be associated with worse COVID-19 outcomes and slower recovery [13]. Some studies did not support these observations [15,24]; however, their results were probably influenced by confounder factors as age and comorbidities.

Takahashi et al. showed that LTBI was associated with a reduction of COVID-19 mortality; in particular, a lower COVID-19 case fatality rate was registered in South East Asia, the region with the highest LTBI burden, in comparison with Europe, where LTBI incidence is five times lower [25]. Another analysis conducted in India on patients with severe COVID-19 showed a correlation between positive QFT and survival rate; at the same time, none of COVID-19 dead patients had a positive QFT [26]. Additionally, Madan et al. evidenced that LTBI patients with COVID-19 had significantly higher lymphocyte and monocyte percentages, lower C-reactive protein, lesser radiologic involvement, and a milder COVID-19 clinical course in comparison with patients without LTBI [27]. Latent tuberculous infection in high TB burden countries seems to have a protective effect on COVID-19 clinical progression, despite conflicting evidence [28].

Our study involved a population living in a low TB prevalence area and with low incidence of LTBI, including 160 patients of Italian nationality and 53 from other countries (18 Central and South America, 17 East Europe, 11 Asia, 7 Africa). Most patients were of

Italian nationality in all three cohorts (Pos, Neg, Seq); in particular, Seq cohort registered the lowest number of foreign patients (11.1%, *p* = 0.023). People with QFT positivity and without radiological sequelae (Pos group) had a comparable COVID-19 in-hospital mortality rate and risk of OTI with respect to patients with negative QFT (Neg group). Patients with TB radiological sequelae (Seq group) showed higher mortality (50%) and risk of OTI (16.7%), regardless of QFT results. These observations are supported by an Italian study involving COVID-19 patients from several countries with active or previous TB, where patients with TB sequelae presented higher COVID-19 mortality, although not statistically significant [11].

The differences observed between low and high TB burden countries could be related to the vaccination with Bacillus Calmette–Guerin (BCG), more common in high TB burden countries. Some studies highlighted a potential protective effect of the BCG vaccine on COVID-19 mortality probably through a stimulation of innate immune memory [29–33]. This hypothesis currently needs more evidence [34].

In our study, the Seq population was characterized by an increased in-hospital mortality (50%, *p* < 0.001) and OTI risk (16.7%, *p* = 0.045). Considering LTBI category as QFT positivity and/or presence of TB radiological sequelae (Pos and Seq groups), COVID-19 in-hospital mortality (39.2%) and OTI risk (13.7%) were significantly higher with respect to the population without LTBI (Neg group). However, these observations are largely due to the higher mortality rate and OTI risk observed in the Seq group, in which the CCI mean value was significantly higher when compared to the other two groups. Multivariate analysis confirmed that both radiological sequelae and CCI are factors independently associated with higher mortality rates. Therefore, despite the higher CCI mean value, we cannot exclude the correlation between COVID-19 in-hospital mortality and presence of radiological TB sequelae.

To the best of our knowledge, this is one of the few studies analyzing the potential correlation between COVID-19 clinical progression and LTBI in a low TB-prevalence country. Several limitations can be evidenced: it was a monocentric and retrospective study with a small size final population due to a limited number of QFT performed and radiological reports available. Moreover, the sample sizes vary significantly between groups in this study, and this might have potentially impacted the statistical analyses and conclusions drawn from the data. Although we used a nonparametric test to address this issue, we are aware of these unequal sample sizes and we tried to interpret the findings accordingly. Additionally, primary and composite outcomes were likely influenced by the high CCI of patients with radiological sequelae: the Seq group consisted mainly of elderly Italian patients who had come into contact with TB during their youth, when TB was endemic in our country. Finally, the exclusion of active TB and other pulmonary diseases relied only on the evaluation of clinical and radiological data by expert radiologists and clinicians, considering the retrospective nature of the study.

## 5. Conclusions

A higher risk of in-hospital mortality was evidenced in patients that were admitted for COVID-19 with LTBI, defined as a positivity to QFT and/or the evidence of radiological sequelae. However, the results are largely influenced by the poor outcome observed in patients with radiological TB sequelae, evidencing a possible correlation between COVID-19 in-hospital mortality and the presence of TB sequelae.

**Author Contributions:** Conceptualization: J.M., A.B. and V.M.; Methodology: J.M. and F.L.; Software: F.L., F.M. and R.P.; Validation: J.M., F.L. and F.M.; Formal analysis: F.L., R.P. and F.M.; Investigation: F.M., F.S., R.P., J.M. and S.T.K.; Resources: F.M., F.S., R.P., J.M., C.M. and T.C.O.O.A.W.; Data Curation: J.M., A.B. and V.M.; Writing—Original Draft Preparation: F.M., F.S., R.P. and J.M.; Writing—Review and Editing: F.L., A.B., C.M. and J.M.; Visualization: F.M. and R.P.; Supervision: J.M.; Project Administration: J.M. All authors have read and agreed to the published version of the manuscript.

**Funding:** This research received no external funding.

**Institutional Review Board Statement:** The study was conducted according to the guidelines of the Declaration of Helsinki, and approved by the Ethics Committee of AOUC Careggi, Florence, Italy ((protocol code 17104_oss, 31/03/2020).

**Informed Consent Statement:** Informed consent was obtained from all subjects involved in the study.

**Data Availability Statement:** The datasets used and/or analysed during the current study are available from the corresponding author on reasonable request.

**Acknowledgments:** COCORA Working Group: Filippo Lagi, Matteo Piccica, Lucia Graziani, Iacopo Vellere, Annarita Botta, Marta Tilli, Letizia Ottino, Beatrice Borchi, Marco Pozzi, Filippo Bartalesi, Jessica Mencarini, Michele Spinicci, Lorenzo Zammarchi, Filippo Pieralli, Giovanni Zagli, Carlo Nozzoli, Stefano Romagnoli, Alessandro Bartoloni, Irene Campolmi, Nicoletta Di Lauria, Giovanni Millotti, Gregorio Basile, Massimo Meli, Pier Giorgio Rogasi, Dario Bartolozzi, Paola Corsi, Marcello Mazzetti, Alberto Farese, Silvia Bresci, Annalisa Cavallo, Michele Trotta, Giampaolo Corti, Alessandro Morettini, Loredana Poggesi, Angelo Rafaele De Gaudio, Adriano Peris, Paolo Fontanari, Paola Parronchi, Fabio Almerigogna, Francesco Annunziato, Francesco Liotta, Lorenzo Cosmi, Alessandra Vultaggio, Andrea Matucci, Manuela Angileri, Alessandra Ipponi, Michele Cecchi, Silvia Benemei, Alessandro Maria Vannucchi, Marco Matucci-Cerinic, Lucia Turco.

**Conflicts of Interest:** The authors declare that they have no competing interest.

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
