# Peer review of "Latent Tuberculosis Infection and COVID-19: Analysis of a Cohort of Patients from Careggi University Hospital (Florence, Italy)"

_2036-7449, doi:10.3390/idr15060068_

Round 1

Reviewer 1 Report (Previous Reviewer 1)

Comments and Suggestions for Authors

I feel that the authors have adequately addressed the issues in the manuscript. 

Author Response

Thank you for your previous suggestions. 

Reviewer 2 Report (Previous Reviewer 2)

Comments and Suggestions for Authors

The authors have improved the manuscript. the manuscrit can be accepted.

Comments on the Quality of English Language

Minor editing of English language required

Author Response

Thank you for your previous comments and suggestions.

Minor editing of English language was done (see highlighted sentences). 

Reviewer 3 Report (New Reviewer)

Comments and Suggestions for Authors

In the manuscript titled "Latent tuberculosis infection and COVID-19: analysis of a cohort of patients from Careggi University Hospital (Florence, Italy), authors  report a retrospective and observational study examining the impact of latent tuberculosis infection (LTBI) on COVID-19 clinical progression. A few case studies have shown that Tuberculosis reactivation results from previous latent bacteria that becomes active either from inducible factors or spontaneously (PMID: 36845786)

Please see my comments below:

1. The abstract could be improved, particularly in the last sentence, to provide a clearer and more concise summary of the study's findings.

2. TB is an opportunistic pathogen, reactivation results from previous latent bacteria that becomes active either from inducible factors or spontaneously. Risk factors for reactivation include HIV/AIDS, steroid use, diabetes, kidney disease, and smoking. This information is important to mention in the introduction. Also did the authors consider these risk factors between the groups?

3. The variation of sample size between the groups is very large. Negative =162, positive =15, sequele=36. Could authors comment on how such big variation should not affect the statistics?

4. Studies have shown that defects or interference of the IFN-γ pathway can cause susceptibility to intracellular infections, including tuberculosis. It has been proposed that there may be an acquired disruption in this pathway caused by COVID-19, this is a crucial information that needs to be added in discussion to explain how COVID might be linked to TB.

Comments on the Quality of English Language

The abstract could be improved, particularly in the last sentence, to provide a clearer and more concise summary of the study's findings.

Author Response

Thank you for your comments and suggestions (see uploaded file).

Minor English editing was done (see highlighted sentences in the manuscript). 

This manuscript is a resubmission of an earlier submission. The following is a list of the peer review reports and author responses from that submission.

Round 1

Reviewer 1 Report

Comments and Suggestions for Authors

Reviewer 2 Report

Comments and Suggestions for Authors

Dear authors, I have had the pleasure of reading your manuscript and I have some observations

1. You write in line 117 that patients with indeterminate QFT were 11 and line 119 write 12, please check it.

2. Line 41 Mycobacterium tuberculosis should write in italic. 

Comments on the Quality of English Language

Minor editing of English language required

Reviewer 3 Report

Comments and Suggestions for Authors

This is an interesting review that describes effects of LTBI on COVID 19 patients. There are three categories of patients in the study cohort of   213 patients and these are QFT positive, QFT negative and radioassay sequelae regardless of the QFT status.  That manuscript draws attention on  an important observation that LTBI population is more prone to COVID 19 morbidity but is written very poorly with numerous grammatical errors and incomplete sentences.  The population index 10000 million is wrong, it is either 10 million or 10,000.   There are numerous errors like this and these need to be edited.  I also think Table 1 not needed

Comments on the Quality of English Language

Manuscript needs to be modified and improved. Editing in sentences needed,  There are incomplete sentences such as 'to detects LTBI'; 'none of COVID-19 died patient's 

These need to be corrected